# Decision Support System to Classify and Optimize the Energy Efficiency in Smart Buildings: A Data Analytics Approach

**DOI:** 10.3390/s22041380

**Published:** 2022-02-11

**Authors:** Manuel Peña, Félix Biscarri, Enrique Personal, Carlos León

**Affiliations:** Electronic Technology Department, School of Computer Science and Engineering, University of Seville, Av. Reina Mercedes S/N, 41012 Seville, Spain; fbiscarri@us.es (F.B.); epersonal@us.es (E.P.); cleon@us.es (C.L.)

**Keywords:** smart building, energy efficiency, data analytics, energy optimization, decision support system

## Abstract

In this paper, an intelligent data analysis method for modeling and optimizing energy efficiency in smart buildings through Data Analytics (DA) is proposed. The objective of this proposal is to provide a Decision Support System (DSS) able to support experts in quantifying and optimizing energy efficiency in smart buildings, as well as reveal insights that support the detection of anomalous behaviors in early stages. Firstly, historical data and Energy Efficiency Indicators (EEIs) of the building are analyzed to extract the knowledge from behavioral patterns of historical data of the building. Then, using this knowledge, a classification method to compare days with different features, seasons and other characteristics is proposed. The resulting clusters are further analyzed, inferring key features to predict and quantify energy efficiency on days with similar features but with potentially different behaviors. Finally, the results reveal some insights able to highlight inefficiencies and correlate anomalous behaviors with EE in the smart building. The approach proposed in this work was tested on the BlueNet building and also integrated with Eugene, a commercial EE tool for optimizing energy consumption in smart buildings.

## 1. Introduction

The growth of energy consumption, energy resource exhaustion and significant environmental impacts [1,2] have raised concerns in most countries, which have entered international agreements for the benefit of society, such as the Paris Agreement in 2015 [3]. With a total energy consumption of 41% [4,5] in residential, public service and commercial sectors, which represents 24% of the world’s CO_2_ emissions [6], these sectors constitute one of the key areas of interest to address, where most action plans are focused on improving Energy Efficiency (EE) through the promotion of renewable energies and also evolving systems to minimize energy consumption. Given this scenario, this work tackles the EE challenge by proposing a new approach to support these initiatives through the optimization of EE in smart buildings using data analytics techniques with the aim of reducing energy consumption and, therefore, reducing CO_2_ emissions.

Because of the relevance of these initiatives for the future of society, many international organizations are sponsoring these initiatives [7,8], such as the International Energy Agency (IEA), the U.S. Department of Energy—Energy Information Administration (DOE/EIA), the Energy European Commission (EEC) and the Organisation for Economic Co-operation and Development (OECD), with directives such as Energy Efficiency Directive 2012/27/EU (EU) or Executive Order (EO) 13514, Federal Leadership in Environmental, Energy, and Economic Performance, October 2009 (DOE/EIA). Additionally, substantial opportunities to improve energy efficiency have been expressed by the International Energy Agency (IEA), as described in [9]. These international initiatives, along with new technologies and their related research proposals, are actively contributing to tackling the above-mentioned challenges. Accordingly, with rising concerns about the exhaustion of energy resources and significant environmental impacts around the world, EE consumption is a rapidly increasing field.

From this perspective, in the last decade, huge efforts have been dedicated to the development of smart cities [10,11,12] in an attempt to design more sustainable cities by applying green energies and a high level of smart services [13,14,15,16,17,18,19,20,21] with the support of new technologies such as IoT [22,23,24,25,26] and 5G [27,28] and the progress made in AI [29,30,31,32,33,34,35,36,37], transport [38], etc. Within this context, efforts have been dedicated to increasing EE in smart buildings with more intelligent behavior, a high level of comfort and environmentally friendly operation [34,37,39,40,41,42,43,44].

There is considerable research focus on energy optimization in buildings. The increasingly sophisticated Building Automation System (BAS) [45] has become the cornerstone of intelligent modern buildings, integrating energy supply and demand factors, often known as Demand-Side Management (DSM), as energy efficiency policies that predict demand, as presented in [46]. Accordingly, most of the progress has been made in Energy Management and Control Systems (EMCSs) [47], which can manage energy policies in real time with the aim of maintaining a high level of comfort with minimum power consumption in different operating conditions [18,48,49]. In EMCSs, one of the most complex problems is optimization according to real-time environmental variables of the building. Current research approaches try to solve this problem from different points of view and with diverse techniques [50] and in different potential scenarios [51,52,53]. These approaches started with basic actions to improve EE [51] until more sophisticated energy management models were introduced, such as those based on a predictive controller in a Supervisory Control and Data Acquisition (SCADA) system [54], multi-objective models through a multi-criteria decision analysis [55,56,57], a more complex approach with Multi-Agent Systems (MAS) [58] or MAS based on occupant behaviors [59], a model based on a Markov Decision Model (MDM) [60] or a predictive control model combined with weather forecasting [61].

Within the above context, the current work is part of the KnoHolEM project, associated with FP7 (FP7-285229 KnoHolEM). The main purposes of this project are mainly focused on achieving the following seven objectives: (1) a functional energy-oriented building model complemented by a corresponding generic building ontology, (2) a specific building behavior model completed by a building-specific ontology, (3) data-mining procedures for detailed real-time energy consumption analysis, (4) algorithms derived from the building-specific ontology running in real time to acquire energy efficiency measures, (5) software for the synthesis and validation of real-time control algorithms, (6) the definition and engineering of hardware and firmware for real-time communication and optimization of energy in buildings and (7) and an interactive virtual reality smart building simulator.

In this sense and following the third and fourth objectives of the KnoHolEM project, the aim of the present paper is to optimize EE in smart buildings through a data analytics approach based on applying Data-Mining (DM) techniques, reducing energy consumption, maintaining a high degree of comfort and being environmentally friendly. Several research efforts have been dedicated to studying energy optimization in buildings with different techniques [62]. The efficiency of a building depends heavily on the way that it is used and how it is managed. In [63], the authors provided an interesting analysis of energy-efficient building design through DM techniques. In [64], the semantic modeling of building systems to support advanced data analytics for EE improvements was described. The authors of [65] presented an advanced DA framework for EE in buildings. However, the analysis of energy consumption at the use stage is essential, and it is clear that the construction characteristics of buildings strongly affect their energy consumption during their life cycle. In [66], the authors presented a review of unsupervised analytics techniques applied to EE enhancement. Finally, [67] proposed a classification approach of energy consumption in buildings.

In this work, a system to classify and measure EE in a smart building is presented. This system is implemented after an analytical process is performed to extract the knowledge hidden in historical building data. In addition, the systems are based on a set of models and algorithms corroborated by EE experts and historical data. This knowledge is used to classify each day analyzed based on its features to then predict the EE based on the insights extracted from historical data and experts. The purpose of this study is the optimization of EE based on measuring and predicting EE for each day from historical and real-time data of the BlueNet building. Subsequently, from the results obtained, it is easy to detect patterns that correlate days with poor EE with a high probability of anomalies. Finally, thanks to the progress made in this field and the results obtained, the system is in the test phase. A future goal is to integrate this framework with commercial EE software called Eugene, owned by the company Isotrol (Seville, Spain).

The paper is organized as follows: In Section 2, the smart building description and data sources of this work are presented. The description of data sources is divided into four subsections: indoor sensors, outdoor sensors, energy analyzers and, finally, EEIs. In Section 3, the proposed method, a data-mining-based DSS to measure and increase energy efficiency in the building, is detailed. It is divided into three subsections: data preprocessing (cleaning, integration, reduction and selection, and transformation), description of the classification module and, finally, the energy efficiency prediction module. In Section 4, the results of the work are presented.

## 2. Case Study: BlueNet Smart Building

Usually, smart buildings are made up of several data-metering units that are managed by an EMCS. This EMCS is able to act as a powerful tool for increasing EE in the building. For EE optimization, it is necessary to know how patterns of energy consumption in the building have been behaving. This knowledge is essential for understanding the consumption patterns of the BlueNet building and, therefore, for optimizing the energy efficiency of the consumption process. Thus, in this section, the BlueNet building and the technology employed in the smart building are described in detail. In addition, a description of data sources used for the classification and prediction of EE in the building is presented.

The BlueNet building is situated in the south of Spain (Seville, 37°24′29.97″ N, 6°00′18.63″ W). This building is used as an office building, and its main activity occurs from 8:00 to 17:30, although there is some variability in the schedule. The data obtained from the BlueNet building are from several data sources, such as indoor sensors, outdoor sensors, energy analyzers and EEIs, covering the main four areas of the building. The indoor sensors are made up of 16 ZigBee sensors per floor that collect indoor variables; ZigBee motes are centralized by a master ZigBee mote used as a gateway that gathers all measurements and sends them to a centralized server. Occupancy data can be obtained by many approaches, such as Passive Infrared (PIR) sensors, ultrasonic systems, camera-based systems, radar systems, CO_2_ sensing, Electromagnetic (EM) signal detection systems, energy measurement devices, computer activity or sensor fusion, chair sensors or the use of multiple technologies to enhance the results [68,69]. In our case, the measurements were obtained through a combination between Radio Frequency Identity (RFID) technology, which is used by proximity cards at the entrance and exit of each room for exhaustive control for staff identity purposes, and Passive Infrared (PIR) sensors incorporated in Zigbee motes positioned strategically throughout the building space. Outdoor sensors comprise a set of sensors placed on the BlueNet building to extract environmental variables and send all of their measurements to a centralized server. Finally, the energy analyzers measure all power consumption variables of the building (HVAC, air mixers, splits, lighting, power plugs and other consumptions). These analyzers are centralized through a master Modbus device. The master Modbus device collects and transmits all consumption measurements to a centralized server that collects all of the BlueNet building information. Besides the information on these areas, a set of useful EEIs and information concerning the schedule of working and non-working days are shared, as this is quite valuable information to understand building consumption patterns and apply the knowledge extracted in this work.

### 2.1. Indoor Sensors

These sets of ZigBee sensors are placed in strategic locations in the BlueNet building and coordinated by a master sensor. The master sensor collects the measurements of the other sensors and sends the data to a master device that acts as a gateway. The purpose of this gateway is to collect sensor information and send it to the BlueNet centralized server. Indoor sensors are programmed to send meter readings every 30 s. These sensors collect the following measurements:Temperature: ZigBee sensors read the temperature in every room of the building. Generally, at least 2 or 3 sensors are used in every room to apply rules that ensure that the correct temperature is obtained.Humidity: ZigBee sensors read the relative humidity in every room of the BlueNet building. It is important to calculate the real feeling of the temperature in each zone.Lux: ZigBee sensors check the real effect of the lighting system by measuring the lux values in every room of the building.Presence: These sensors obtain data that indicate the presence of all occupants in the BlueNet building by identifying every person with a unique id and obtaining information about the time that every person is in BlueNet building rooms.

### 2.2. Outdoor Sensors

In addition, there is a set of sensors that collect metering data of the BlueNet environment. These sensors are responsible for collecting environmental data. Outdoor sensors are programmed to measure metering data every 10 min. The main measurements are:Temperature: These sensors take the environmental temperature every 10 min and are also able to provide the maximum, minimum and mean temperature of each day.Humidity: The outdoor sensors measure the environmental humidity and the amount of rain fallen to calculate the feeling of environmental temperature.Sunshine: These sensors obtain the amount of sunshine that irradiates onto the building every day.Wind: The outdoor sensors also obtain the amount and direction of the wind in the building environment.

### 2.3. Energy Analyzers

Energy analyzers are placed in an energy distribution panel inside of the BlueNet building. They are connected to a Programmable Logic Controller (PLC), which acts as a master Modbus device of the energy analyzers. These analyzers meter the four typologies of energy consumption in the building (HVAC, lights, power plugs and other consumptions). These analyzers are programmed to send measurements every 5 min to the master Modbus device. The principal features of every typology of consumption are described below.

#### 2.3.1. HVAC

In the BlueNet building, HVAC is based on a VRV system. This system is made up of a set of indoor units (Daikin FXSQ-M7V1B, Daikin AC Spain S.A, Madrid, Spain) and a set of outdoor units (Daikin RXYQ-MY1B VRV II inverter—Daikin AC Spain S.A, Madrid, Spain—with heat pump). All units are connected by a DIII-Net. These connections are centralized in a DMS504B51 Daikin Lonworks Interface, which is in turn connected to a centralized server through a Lonworks/Modbus gateway (IntesisBox—HMS Industrial Networks AB S.L.U., Barcelona, Spain) using the communication protocol I3E.

HVAC was the primary area of consumption in this study due to its strong influence on EE [50,51,52]. Specifically, HVAC systems are appliances with the largest consumption in the building and are also the most controllable. HVAC management has the largest margin for EE improvement, consuming 143,876.7 MWh in 509 days, which represents 40.11% of the total energy consumption in the building.

#### 2.3.2. Lights

The BlueNet building is made up of a luminary system, Philips Light Master (Luminary with ballast HF-R TD 318), connected through a DALI bus. This bus is centralized and managed by an LRC5141 controller, and it is connected to a centralized server through a Lonworks/Modbus gateway (IntesisBox—HMS Industrial Networks AB S.L.U., Barcelona, Spain). This server is responsible for sharing the data and applying certain energy management policies.

The lighting only accounted for 19.03% of total energy consumption in the BlueNet building, consuming 68,281.7 MWh throughout the 509 studied days. Thus, it is difficult to reduce this consumption (lighting is a necessary building function, and it is associated with occupancy in rooms). Nonetheless, the study of this field provides interesting information on EE, such as occupation patterns, anomalous consumptions or group behaviors.

#### 2.3.3. Power

The consumption at power points in the BlueNet building is mainly the result of ICT equipment (PCs, servers, media, etc.). The majority of power consumption usually occurs during working hours, with the exception of some services that provide support 24 h a day.

This area is the second most relevant for this study, accounting for 120,368.7 MWh of the total consumption in the study period. Although it constitutes the second largest consumption in the building, accounting for 33.56% of the total, this energy consumption is hard to improve from the EE point of view because computers perform scheduled tasks outside of working hours, making it difficult to reduce this type of energy consumption.

#### 2.3.4. Others

This area accounts for the minority of energy consumption, with only 26,170 MWh of total consumption. In addition, it has a strongly fixed consumption that is difficult to manage. Thus, with only 7.3% of total energy consumption, it is the least relevant field in EE.

### 2.4. Energy Efficiency Indicators (EEIs)

A set of EEIs of the BlueNet building were analyzed. These EEIs are based on experts’ knowledge and historical data behavior of each relevant area in the BlueNet building, as described in [21]. The EE behavior in the BlueNet building is evaluated on the basis of these EEIs. In addition, each EEI provides the knowledge required to detect EE behaviors and anomalies. The EEIs are described further below.

#### 2.4.1. Operational Changes in HVAC Compressor (OCC) Indicator

This EEI counts the number of daily on–off operations in the compressor. A large number of daily on–off operations are considered anomalous or inefficient, and it can cause one of the largest energy leaks and high inefficiency, greatly increasing energy consumption. A high on–off operation variance could indicate a possible anomaly in HVAC management (the HVAC is poorly dimensioned for this room, the HVAC is too powerful for this room, or there is a possible malfunction in the temperature sensor). Moreover, a compressor with a high OCC is prone to break down and have a shorter lifetime.

#### 2.4.2. Number of Operational Regime Changes in the HVAC Compressor (ORCC) Indicator

This EEI counts the number of daily ORCC periods and the number of minutes in which the daily ORCC periods occurred. An ORCC is defined as a change in the compressor power consumption greater than 0.5 kW with respect to the previous measurement (10 min). These parameters were specified based on the results of DM techniques under the consensus of Isotrol HVAC experts. Thus, a large number of operational regime changes in the HVAC compressor (ORCC) is considered abnormal or inefficient.

This EEI can indicate that the HVAC system is not properly calibrated (HVAC is too powerful for this requirement) or that the temperature in the room is forcing the HVAC compressor to constantly change its operating mode. Furthermore, this EEI can denote a possible improvement in HVAC management, softening the HVAC consumption curve.

#### 2.4.3. Switch on HVAC Compressor and Abnormal Changes in Indoor Temperature (SONCCIT) Indicator

This EEI counts the number of total minutes per day with a SONCCIT anomaly. In addition, this EEI averages the daily active power in the HVAC compressor during the compressor anomaly and the number of periods per day in which a switched-on compressor anomaly is observed (an anomalous period is specified as the aggregation of consecutive anomalous data points). A SONCCIT anomaly is defined when the HVAC system is turned on (the HVAC compressor consumption is higher than 1.7 kWh) and produces a change in indoor temperature greater than or equal to 1 °C between samples (10 min).

It is considered abnormal or inefficient when the room temperature decreases sharply (winter) or increases sharply (summer) while the compressor is running. This could be due to a sudden leakage of heat (winter) or a sudden influx of heat (summer) in the room, which counteracts the effect of the HVAC system (i.e., opened windows or doors).

#### 2.4.4. Switch off HVAC Compressor and Abnormal Changes in Indoor Temperature (SOFFCCIT) Indicator

This EEI counts the total minutes per day with a SOFFCCIT anomaly. In addition, this EEI averages the daily active power in the HVAC compressor and the number of periods per day in which a switched-off compressor anomaly is observed (as in SONCCIT, an anomalous period is defined as the aggregation of consecutive anomalous data points).

It is considered abnormal or inefficient when the room temperature rises sharply (winter) while the compressor is not running. This could be due to a heat source (electric heater) replacing the HVAC system and can indicate the inefficient use of power energy. From historical data, a 1 °C increase between samples (10 min) during the winter period or a 1 °C decrease during the summer months is considered anomalous.

#### 2.4.5. No Persons in BlueNet Building and Switch on HVAC Compressors (NPSONC) Indicator

This EEI counts the total daily minutes in which an NPSONC anomaly is detected, the number of periods per day with an NPSONC anomaly (an anomalous period is defined as the aggregation of consecutive anomalous data points) and the average active power consumption per day by the HVAC compressor during the anomaly. An anomalous function of the compressor is identified when there is an absence of occupants or the lights are switched off and the compressor is switched on (NPSONC). This could indicate that the air conditioner is switched on accidentally, considering that there are no building occupants if the lights are not switched on.

In this study, five data sources were analyzed: indoor sensors, outdoor sensors, schedule of working days, energy analyzers and EEIs. These data sources were employed during this work, in which every area was exhaustively and carefully examined to detect any possible improvement in energy management.

Some important information for this first analysis of historical data behaviors yielded the following results: 40.11% of building consumption was attributable to HVAC, 33.56% of energy consumption was due to power, 19.03% was spent on lighting, and another 7.3% was due to other activities, with an accumulated energy consumption of 358,697.1 MWh in the BlueNet building during the analyzed period. The most relevant consumption of the BlueNet building was due to HVAC operation with 143,876.7 MWh, and specifically, the major HVAC consumption was attributable to the compressor motor engine consumption, accounting for 46.38% of the total HVAC consumption.

Thus, a system to measure and optimize EE in the BlueNet building through DA techniques was developed. The objectives of this system were to: extract the knowledge hidden in BlueNet building data, develop a classification module and build an EE prediction module that helps to predict EE for each day. This EE classification is able to compare EE on days with similar characteristics, regardless of the season and other factors, which would be difficult to compare without this information, and also allows the presence of anomalous energy consumptions and other possible problems to be identified. The EE prediction module is able to quantify the energy efficiency every day, comparing days with similar energy efficiency conditions based on clusters.

The architecture of the EE optimization system is depicted in Figure 1.

## 3. The Data-Mining-Based Decision Support System to Optimize EE in the Smart Building

This section describes the approach selected to optimize the EE in the BlueNet building. This approach is based on a hybrid decision support system that combines a search based on the historical data to classify the energy consumption on days with similar features and a prediction module for additional EE interpolations. This approach takes advantage of the knowledge extracted from the historical data of the building, making it possible to reveal information about behavioral patterns in energy management, as well as knowledge provided by HVAC experts to optimize EE. Therefore, this approach comprises a module for the classification of energy consumption, which is based on CR&T decision tree [70] and cluster classification, and a module for EE prediction, which is based on metrics. With both modules, the days are analyzed with the EEI results to discover patterns and detect anomalies or other possible problems.

Data can provide some insights hidden in the behavior of historical data. Normally, the expert’s knowledge is hidden in the collected dataset. Knowledge Discovery in Databases (KDD) refers to the overall data-mining process of discovering useful knowledge from large amounts of data. The DM process consisted of 6 essential phases: understanding the business, understanding the data, data preprocessing, modeling, evaluation and deployment. Once the phases of understanding the business and data were carried out, the next phase was the preprocessing of data. In this phase, the data were cleaned, and the different data sources were integrated, reduced and selected, and finally, transformed [71,72] (described in Section 3.1). After these phases, data were prepared for the modeling phase, which is explained in Section 3.2 and Section 3.3 and evaluated in Section 4.

On the one hand, the first study aimed to extract the knowledge of energy consumption behaviors of the BlueNet building and determine how the energy has been consumed (e.g., regime changes in the compressor, operational changes in the compressor, lighting patterns and others). Secondly, the influence of each BlueNet building variable in every area (indoor sensors, outdoor sensors, HVAC, lighting, power and other consumption measurements, and EEIs) was quantified.

On the other hand, after extracting the knowledge hidden in BlueNet building data, an EE classification of the historical data was carried out through a hybrid system of classification. This EE classification allows the establishment of a relationship between days with similar features and also enables experts to compare the behavior on these days in EE terms. The classification module is based on a decision tree, the clustering of historical building data and the use of a set of energy efficiency indicators (Section 3.2). As a result, this classification identifies the presence of anomalous energy consumptions and other possible problems. In addition, a prediction system to quantify EE every day was developed (Section 3.3). This prediction is based on the features of each cluster classification, with the aim of assisting in the quantification of energy efficiency every day.

In order to carry out this work, a powerful and extended tool in the analytics area was used, namely, the SPSS Modeler tool (originally Statistical Package for Social Sciences Inc., an IBM company). This tool was used to perform the preprocessing and modeling tasks, as well as to evaluate the models. In addition, SPSS Modeler includes IA libraries used by the classification and prediction systems through DM techniques.

Currently, the decision support system to optimize EE in the BlueNet building is in the testing phase, and it will be connected to Eugene, an EE tool owned by Isotrol. These modules provided all of the knowledge extracted from the large amount of data provided by the BlueNet building with the aim of improving the results of the EE tool.

### 3.1. Data Preprocessing

The DM process requires an initial phase of data preprocessing, in which the data are analyzed, filtered and formatted [71]. BlueNet data comprise different data sources: energy consumption, environmental sensors inside of the building, external climate sensors, EEIs and other sources of data, all of which have their own temporal frequency. Thus, to manage different timestamps among the recorded data, the frequency was synchronized with a period of 10 min. The time interval for these data sources is between January 2011 and March 2013 (509 days).

The type of data strongly depends on the source of the data. The types of data and their time bases are as follows:
-Indoor sensors (30 s basis): mote_id, timestamp (YYYY/MM/DD hh:dd:ss), temperature (Celsius degrees), percentage_humidity, CO_2_ and lux (lumens).-Indoor sensors (30 s basis): mote_id, timestamp (YYYY/MM/DD hh:dd:ss) and employee_id (presence).-Outdoor sensors (10 min basis): sensor_id, latitude, longitude, timestamp (yyyy/mm/dd hh:dd:ss), wind_direction (degrees), max_wind_speed (m/s), min_wind_speed (m/s), ave_wind_speed (m/s), UV_index, max_humidity, min_humidity, ave_humidity, precipitation (l/$m^{2}$) and sunshine_radiation (W/$m^{2}$).-Energy Analyzers—HVAC (5 min basis): timestamp (YYYY/MM/DD hh:mm:ss), AP_CLI_FASE1 (kW), AP_CLI_FASE2 (kW) and AP_CLI_FASE3 (kW).-Energy Analyzers—Lights (5 min basis): timestamp (YYYY/MM/DD hh:mm:ss), AP_LIG_FASE1 (kW), AP_LIG_FASE2 (kW) and AP_LIG_FASE3 (kW).-Energy Analyzers—Power (5 min basis): timestamp (YYYY/MM/DD hh:mm:ss), AP_POW_FASE1 (kW), AP_POW_FASE2 (kW) and AP_POW_FASE3 (kW).-Energy Efficiency Indicators—OCC, ORCC, SONCCIT, SOFFCCIT and NPSONC (10 min basis): timestamp (YYYY/MM/DD hh:mm:ss) and anomaly (true/false).-Energy Efficiency Indicators—OCC (10 min basis): timestamp (YYYY/MM/DD hh:mm:ss) and anomaly (true/false).-Energy Efficiency Indicators—ORCC (10 min basis): timestamp (YYYY/MM/DD hh:mm:ss) and anomaly (true/false).-Energy Efficiency Indicators—SONCCIT (10 min basis): timestamp (YYYY/MM/DD hh:mm:ss) and anomaly (true/false).-Energy Efficiency Indicators—SOFFCCIT (10 min basis): timestamp (YYYY/MM/DD hh:mm:ss) and anomaly (true/false).-Energy Efficiency Indicators—NPSONC (10 min basis): timestamp (YYYY/MM/DD hh:mm:ss) and anomaly (true/false).

In this regard, the techniques used during the entire preprocessing stage were the following:-Data cleaning: removing all missing and null values, as well as inconsistencies found in each data source;-Data transformation: normalizing all data to the same period (10 min) to facilitate the aggregation of data and their analysis;-Data reduction: simplifying the data with the aim of providing meaningful data.

Once the data were normalized and standardized, the data sample consisted of 5,088,765 records corresponding to EEI data and the set of variables of the BlueNet building. From all of these variables, only the most relevant ones were selected for use in the study. A list of these variables is summarized in Appendix A.

Initially, the 5,088,765 records of the dataset were analyzed. From these records, 636,848 missing data records were found, corresponding to null values. These records were filtered and removed from the data sample. After the exclusion of outliers, the next requirement was ensuring the data quality. Thus, a set of rules were established to guarantee the data quality:Detect every outlier in the data sample and fix it with the average value of data dispersion through DM techniques.Data consistency validation: every sample of data requires at least one value for each 30 min period.

As a consequence of this first phase of data quality control, 636,848 records were filtered in the preprocessing task, corresponding to null values. After applying basic rules to ensure data consistency, 34 records were filtered. Finally, the data sample was reduced from 5,088,765 records to 4,451,883 records.

Once the preprocessing phase was carried out, a set of 25 relevant variables was selected from the total of variables measured in the BlueNet building for the tasks of data classification and data prediction (described in Appendix A). This selection was carefully analyzed by studying the degree of the influence of each selected variable through Principal Component Analysis (PCA) techniques [73]. At first, the standardization of continuous variables was carried out, so each one contributed equally to the analysis. Second, a covariance matrix was established to identify correlations between the selected variables; pairs that had a positive score were correlated, and pairs that had a negative score were inversely correlated. Third, the eigenvectors and eigenvalues of the covariance matrix were computed to identify the principal components. These principal components are new variables constructed as linear combinations or mixtures of the initial variables; principal components are uncorrelated and provide the maximum information based on their variance. On the basis of the results, the most meaningful variables were selected, providing 25 relevant variables. Most of the variables analyzed were used in other similar research studies [74,75].

### 3.2. Classification Module

The classification module was developed through DM techniques with the aim of classifying days with similar features and comparing these days to quantify the EE. These clusters were developed with the aim of extracting insights about how different features affected days in EE terms. The second aim of these comparisons is to determine the clusters in which the detected anomalies are concentrated and infer key features that can provide some insights to reveal anomalies. Once the classification and detection are carried out, the main objective is to predict days with anomalies and then apply policies to improve the EE. This classification module uses the most influential information supplied by EEIs; HVAC, lighting, indoor and outdoor environmental information; BlueNet occupation; and holiday/work schedule.

In the first instance, data for the classification module were analyzed through Generalized Rule Induction (GRI) [76] and the Apriori algorithm [77] to study the correlation between the selected variables and consumption. GRI begins with the original set *S* as a root node. It iterates through every unused attribute of the set *S* and calculates the entropy *H(S)* or the information gain *IG(S)* of that attribute. It selects the attribute with the smallest entropy (or largest *IG*), and the set *S* is partitioned by the selected attribute. Finally, the algorithm continues recursion on each subset.

Entropy *H(S)* is a measure of the amount of uncertainty in the dataset *S:*$$\;\;\;\;\;H\left(S\right)=\sum_{x\in X}p\left(x\right)log_{2}\;p\left(X\right)$$
where *S* is the current dataset for which entropy is being calculated, *X* is the set of classes in *S,* and *p(x)* is the proportion of the number of elements in class *x* to the number of elements in set *S.*

Information Gain *IG(A)* is the measure of the difference in entropy from before to after the set *S* is split on the basis of an attribute *A*. In other words, it measures how much uncertainty in *S* was reduced after splitting set *S* on the basis of attribute *A*:$$IG\left(S,A\right)=H\left(S\right)-\sum_{t\in T}p\left(t\right)H\left(t\right)=H\left(S\right)-H(S|A)$$
where *H(S)* is the entropy of set *S*; *T* is the subsets created by splitting the set *S* by attribute *A*, such as $S={⋃}_{t\in T}t$; *p(t)* is the proportion of the number of elements in *t* to the number of elements in set *S*; and *H(t)* is the entropy of subset *t.*

Additionally, the Apriori algorithm is used for frequent item set mining and association rule learning, and it uses breadth-first search and a hash tree structure (Algorithm 1).
**Algorithm 1.** A Priori with breadth-first search.*Apriori(T,*ε*)*       $L_{1}$ ⟵ {*large* 1 − *itemsets*}      *k*
⟵ 2      *while*
$L_{k-1}\neq \emptyset$         $C_{k}⟵${*a* ∪ {*b*} | *a*
∈ $L_{k-1}$
∧
*b*
∉
*a*} − {*c*|{*s*|*s*
⊆
*c*
∧ |*s*| = *k* − 1}$⊈L_{k-1}$*}*        *for transformations*
t∈T           $C_{t}$ ⟵{*c*|*c*
∈ $C_{k}$ ∧
*c*
⊆
*t*}          *for candidates*
$c\in C_{t}$            *count*[*c*] ⟵*count*[*c* + 1]         $L_{k}$ ⟵{*c*|*c*
∈ $C_{k}$ ∧
*count*[*c*] ≥ ε}        *k*
⟵*k* + 1      *return*
⋃kLkwhere*T*—The set of data;ε—Confidence threshold;k—Size of the set of candidate items;Ck—Candidate set at level *k*;*c*—Candidate *c*;*count*[*c*]—Pointer to the candidate set *c*.

As a result, 24 features were selected from the initial set of 25 variables that were previously filtered. These features provided more detailed and useful information to develop the classification module through DM techniques with the aim of supplying the most accurate results possible. The features selected for the classification module are shown in Table 1.

Subsequently, the resultant variables, also called features, were analyzed and classified according to outdoor temperature (AEMET_AT), occupant presence (PRESENCE_ID_ENT and PRESENCE_ID_EXI) and HVAC compressor consumption (AP_COMPRESSORS_MEAN).

Firstly, a filter was applied to exclude all data without compressor measurements. On the one hand, after obtaining the sample without null compressor function values, an analysis of this sample was carried out to realize the segmentation of the data according to outdoor temperature (AEMET_AT). As a result of applying a binning algorithm, 8 groups of outdoor temperature (TE_Mean_BIN) were defined with a range from 3.56 °C to 39.52 °C in 5 °C intervals, as is shown in Table 2. The binning algorithm is used to reduce the effects of minor observation errors, and it carried out bucketing, where bins have an equal frequency of 5 degrees Celsius following the formula:(1)$$L=\frac{\max\left(x\right)-\min\left(x\right)}{n}$$
where *L* is the length of the bucket, and *n* is the number of buckets.

On the other hand, the distribution of the building occupancy was analyzed by applying statistical methods and defining 3 clear groups: a group with fewer than 50 persons, which mainly represents holidays; a group with between 50 and 110 persons, which indicates days with medium occupancy; and finally, a group with a range between 110 and 160 persons corresponding to working days with high occupancy, which contains the majority of the sample, as is shown in Figure 2.

Finally, the data were represented in terms of compressor consumption (AP_COMPRESSORS_MEAN), external temperature (AEMET_AT) and occupancy (PRESENCE_ID_ENT and PRESENCE_ID_EXI) to be further analyzed. Furthermore, a discriminant analysis between workdays and non-workdays was performed to provide more depth to our model. The sample distribution is illustrated in Figure 3.

After representing and studying the data distribution through the different features, a classification model based on a C&RT decision tree using Gini impurity measures was carried out, with the aim of modeling the sample based on the mean daily active power of the compressors (AP_COMPRESSORS_MEAN). As a result, a set of rules were obtained. C&RT is used for both classification and regression, and it uses the Gini Index (*GI*) criterion to split a node into subnodes. It starts with the training set as a root node, and after splitting the root node in two, it splits the subsets using the same logic recursively until it finds that further splits will not result in any pure subnodes or reaches the maximum number of leaves in a growing tree. The Gini Index is expressed as follows:$$GI=\overset{c}{\underset{i=0}{\sum }}P_{i}\left(1-P_{i}\right)$$
where *c* is total classes, and$\;P_{i}$ is the probability of class *i*.

This set of rules was the basis for substantiating our clustering model. The scheme of this classification algorithm is shown in Figure 4.

As can be seen at the first level of the decision tree, the sample was classified into 2 groups distinguished by external temperature (*AEMET_AT*). On the one hand, groups with lower external temperature ranging from 3.5 °C to 23.5 °C correspond to groups 1, 2, 3 and 4. On the other hand, groups with higher external temperature ranging from 23.5 °C to 39.5 °C correspond to groups 5, 6 and 7. Once this first classification was carried out, it was possible to observe a division between days with lower temperature: one group has external temperature ranges from 3.5 °C to 18.5 °C (1, 2 and 3) with very little compressor use, and another group has a comfortable temperature range ranging from 18.5 °C to 23.5 °C (4) with slight compressor use.

These 3 groups were derived from the following clusters analyzed to study EE in the BlueNet building.

#### 3.2.1. Cluster 1

In cluster 1 (Figure 5), it was possible to observe a classification of days with lower external temperature ranging from 3.5 °C to 18.5 °C. This group (158 days) had an average compressor consumption ranging from 2.566 kW/h to 2.583 kW/h on working days with an effect of −0.006 and 0.011, respectively. On non-working days (*LaborFestive* equals 1), the average consumption of the compressor ranged from 2.067 kW/h to 2.409 kW/h with an effect of −0.18 and 0.114, respectively, which indicates the effect on the entropy—average—when removing a value from the cluster. Furthermore, it should be noted that most of the days followed this distribution, while scattered days usually coincided with individual cases of high consumption over a short period of time. In addition, non-working days had a homogeneous distribution with compressors having quite low average active power consumption, very close to 2 kW/h.

#### 3.2.2. Cluster 2

In cluster 2 (Figure 6), it was possible to observe a classification of days with a comfortable external temperature that ranged between 18.5 °C and 23.5 °C. This group (38 days) had an average compressor consumption of 3.086 kW/h on workdays with an effect of −0.016. On non-working days, the compressor average was 2.934 kW/h with an effect of −0.136. In addition, it was possible to observe a wider dispersion and high similarity between workdays and non-workday cases. This is caused by a comfortable external temperature ranging between 18.5 °C and 23.5 °C. In addition, in Figure 6, it is possible to observe the number of minutes in which the HVAC system was switched on (*C_FunctionMins*).

#### 3.2.3. Cluster 3

After analyzing cluster 1 and cluster 2, cluster 3 was the most relevant and interesting, and it covered the majority of cases with high external temperature. This cluster, with 116 days, has a somewhat complex distribution because it was analyzed carefully and split into 3 well-defined groups, as is shown in Figure 7.

##### Cluster 3.1

Cluster 3.1 contains the majority of non-working days, in which the compressor had a low range of average energy consumption, with values between 1.7 kW/h and 3 kW/h for the entire period covered in the data sample (26 days) (Figure 8).

##### Cluster 3.2

Cluster 3.2 includes the majority of working days during the year 2011 until the months of July–August (22 days), during which consumption was higher, with values of 3 kW/h and 5 kW/h, except for a day on which data were dispersed to almost 8 kW/h. As it is possible to observe, this cluster corresponds to a season with more moderate temperatures, as is illustrated in Figure 9. Furthermore, in the illustration in Figure 9, it is possible to observe the number of minutes in which the HVAC system was switched on (*C_FunctionMins*), providing further detail.

##### Cluster 3.3

Finally, cluster 3.3 corresponds to most of the working days during 2012 from April to October. This cluster (68 days) is characterized by days with a large dispersion, caused by a hotter season with higher temperatures, and an average compressor consumption ranging between 3 kW/h and 7 kW/h in a regular manner, as is illustrated in Figure 10.

In this case, the behavior of all days is characterized by a high compressor consumption and a high number of changes in compressor consumption. Thus, on days with similar environmental characteristics, the features that indicate the grades of efficiency comprise lower compressor consumption, uniform compressor consumption and a small number of slight changes in compressor consumption.

### 3.3. Energy Efficiency Prediction Module

Once the developed classification module was implemented, the results of this module were carefully studied. As a result of the study, an energy efficiency prediction module was developed. The purpose of this module was mainly to obtain a metric of EE for this study. Therefore, clusters were studied independently, and the EE estimation was based on historical behavior inferred through statistical methods.

First, every cluster was selected, and a distribution analysis for each sample was carried out. The distribution analysis was based on an energy consumption histogram. This distribution curve follows a distribution that is similar to a Gaussian distribution; thus, the EE categories were fitted with a function of the sample distribution, as is shown in Figure 11.

Thus, the days were categorized into 5 EE categories based on cluster dispersion. These categories were established on the basis of the normal distribution, which approximates a Gaussian distribution. The limits for each category comprise the following segments: μ− 3/2 σ, μ− 1/2 σ, μ+ 1/2 σ and μ+ 3/2 σ, where μ is the mean, and σ is covariance. As a result, the consumption on each day was categorized according to every cluster characteristic following the classification based on the data distribution shown in Figure 12.

Besides EE categories for each cluster, this module is able to provide some insights that can help to reveal and detect the patterns and anomalies observed on inefficient days. For example, the consumption behaviors on 4 October 2012 (cat. 1) and 20 September 2012 (cat. 5) had the same features (cluster 3.3), which comprise the same range of outdoor temperatures, number of persons and other environmental conditions, but the energy consumption behavior was very different, as it was highlighted through the EE category carried out by the EE prediction module. Figure 12 shows a wide range of consumption (0–6 kWh vs. 0–12 kWh), a greater number of stop points (5 OCC vs. 3 OCC) and regime changes (43 ORCC with 293 min of anomalous function vs. 77 ORCC with 532 min of anomalous function), a less softened energy consumption curve with a larger variation in energy consumption changes at the same time, etc. All of these results can indicate an anomaly, such as HVAC breakdown, poorly dimensioned HVAC, a possible isolation problem in the room and other issues.

In summary, the EE prediction module indicates the EE category for every day as a function of the historical behavior for the cluster sample. In addition, the EE prediction module, together with the classification module, can indicate possible anomalous consumption patterns that occur in the building and other possible anomalies. These cases were studied, and the results show a tight correlation between the detected OCC and ORCC anomalies and the index of EE, as shown in Table 3, Table 4, Table 5, Table 6 and Table 7. At the same time, it is observed that SOFFCCIT, SONCCIT and NPSONC anomalies are not correlated with inefficient consumption behavior.

The above-mentioned correlation between detected OCC and ORCC anomalies versus the index of EE highlights not only the number of anomalies but also the period (in minutes) of each anomaly for ORCC and the number of stop points for OCC, as shown in Table 8. This insight is even more relevant when one of the principles of EE consists in softening the consumption curve, aiming to avoid peaks and large changes in consumption. Furthermore, OCC and ORCC anomalies denote large changes and peaks in consumption, which means that the HVAC system is poorly dimensioned or calibrated, the HVAC system is too powerful, or a possible breakdown in a temperature sensor has occurred. On the other hand, a compressor with high OCC is prone to break and to have a shortened lifetime. Table 8 highlights the correlation between an increasing number of stop points for OCC and an increasing number of minutes with ORCC in each period when EE decreases. Additionally, in Table 9 are detailed the average of OCC and ORCC anomalies per day and its correlation with each EE Category.

## 4. Summary and Experimental Results

The objective of the present work was to present a decision support system to optimize Energy Management and Control Systems (EMCSs) in smart buildings for any energy-consuming operations. Thus, in order to achieve the main purpose of this work, a model to extract knowledge to realize the efficient management of energy consumption was developed. This model was based on a classification module and a prediction module developed through DM techniques and statistical inference. All BlueNet variables were considered for this EE classification study (indoor, outdoor, energy analyzers, EEIs and work schedule). In addition, the knowledge extracted from data and EE experts was considered for this purpose. Once the study was carried out, five clusters were identified. Each cluster was defined by the following set of conditions:Cluster 1: This cluster represents days with lower external temperature ranging from 3.5 °C to 23.5 °C. This cluster does not discriminate between energy consumption or between work and non-workdays.Cluster 2: This cluster represents days with a softer curve of intermediate temperature ranging from 23.5 °C to 28.5 °C. This cluster does not discriminate between energy consumption or between work and non-workdays.Cluster 3.1: This cluster groups days with higher external temperature ranging from 28.5 °C to 39.6 °C, in which most of the days are non-working days with low energy consumption.Cluster 3.2: This cluster represents days with higher external temperature ranging mainly from 28.5 °C to 33.5 °C. Most of the selected days are working days with high energy consumption.Cluster 3.3: This cluster groups days with higher external temperature ranging mainly from 28.5 °C to 39.6 °C, in which most of the selected days are working days with high energy consumption.

Once clusters were defined, days that belong to the same cluster were compared to observe differences among behaviors and, consequently, the effects of the different variables on each day. In order to validate and quantify the results, the results were analyzed and corroborated by EE experts of the BlueNet building, and additionally, a module for EE prediction was developed. The aim of the EE prediction module was to evaluate EE according to the type of cluster analyzed. The results obtained were applied to the sample of 509 days with the following results.

All of the results in this study were corroborated by EE experts. In summary, only 3.25% of the total days classified were considered to be very efficient, 29% were efficient and 41.36% were classified with normal EE. The remaining 20.29% of the days were considered to be energy inefficient, and 6.36% of days were classified as very inefficient, indicating a large margin for EE improvement, as detailed in Table 10. 

Moreover, in this study, days that represent anomalous behavior were detected with the support of the classification module and prediction module. These anomalous behaviors usually correspond to days that were very inefficient. In addition, these anomalous behaviors can indicate possible problems that are affecting the BlueNet building (HVAC breakdown, incorrectly dimensioned and other problems), as shown in Table 2, Table 3, Table 4, Table 5, Table 6, Table 7 and Table 8. With the results in this study, system integration with EE software could yield substantial benefits: detection of anomalous behaviors in energy consumption, facilitation of energy savings, large profits due to consumption reduction and environmentally friendly management of the building.

## 5. Conclusions

After a bibliographical review, an intensive field of research with strong interest in EE was identified. This review revealed that similar approaches have been considered, but no other works have been performed with the aim of rendering support in EE classification and prediction. Specifically, systems able to measure EE and support the detection of anomalies have not been investigated with similar approaches or results.

This paper presents a system for the optimization of energy consumption in smart buildings. The main purposes of this work are the following: extract the knowledge hidden in building data, develop a classification module and build an EE prediction module that helps to predict EE for each day. This proposal was tested in the BlueNet building scenario, but it is applicable to any other building and is not specific to BlueNet. 

The process of energy optimization was carried out through a hybrid system whose cornerstone is a classification module and an EE prediction module. The classification module is made up of a hybrid system based on a CR&T decision tree and a clustering model. As a result, this module is able to compare EE on days with similar characteristics, regardless of the season and other factors, which are difficult to compare without this information, and it is also able to highlight the presence of anomalous energy consumptions and other possible problems. This module provides an objective point of view that is key to measuring, comparing and predicting the EE for each day.

The EE prediction module is able to quantify the energy efficiency of each day, comparing days with similar EE conditions supported by previous cluster classifications. This module is able to measure and predict the efficiency for each day based on the knowledge extracted from historical data by applying statistical analysis. Furthermore, this module is able to unveil insights that highlight correlations between inefficiencies and anomalous behaviors.

Finally, this work presents a classification for each day of the historical data and their respective EE categories. These results were compared and corroborated by experts to, first, understand how the energy consumption is behaving and, second, understand the reasons for this behavior and how to enhance the efficiency. Furthermore, our approach highlights some evidence that days with less efficiency (6.36% and 20.29% of days respectively) contain more anomalies and that these anomalies also occurred over a greater amount of time. Furthermore, this work unveils an exponential correlation between OCC and ORCC anomalies and EE.

Based on the results obtained in the BlueNet building and aligned with this research area, interesting future research lines that could be explored include the automation of the full process through AI techniques that are able to classify and predict EE in an analytical manner in real time and, based on anomalous behavior, determine how to apply actionable measures to correct and improve EE in the building automatically. On the other hand, this work aimed to support the gap in the understanding of building behavior, as cited in [39]. A future goal is the integration of this module with the commercial EE software called Eugene, whose owner is the company Isotrol.

## Figures and Tables

**Figure 1 sensors-22-01380-f001:**
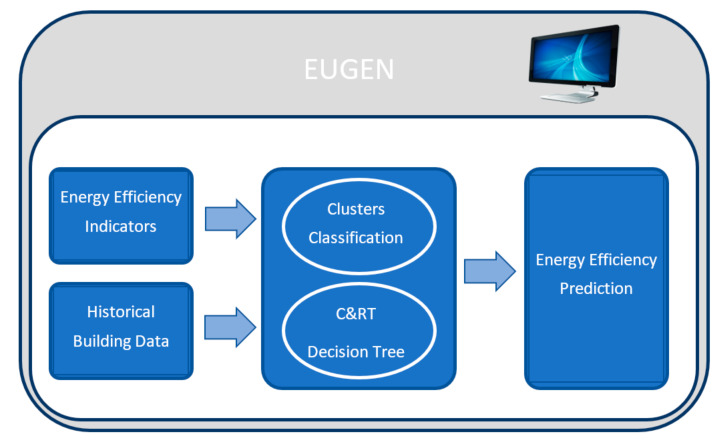
Prototype architecture.

**Figure 2 sensors-22-01380-f002:**
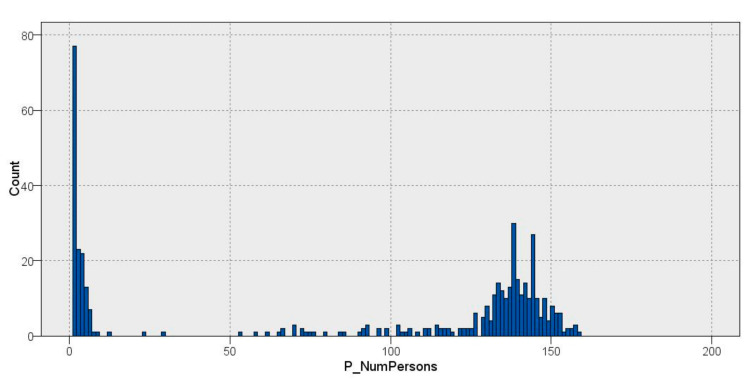
Occupancy distribution.

**Figure 3 sensors-22-01380-f003:**
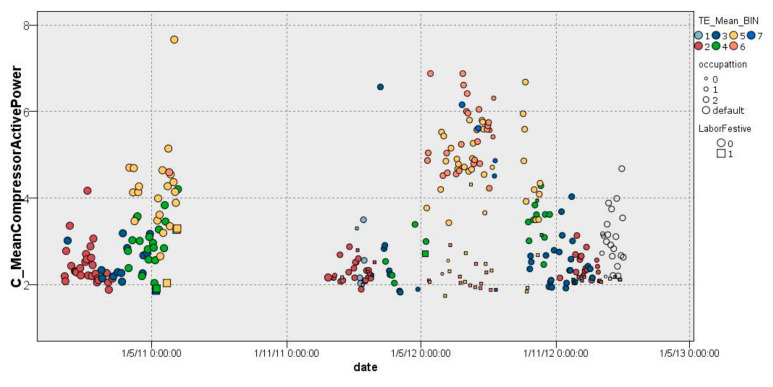
Sample compressor consumption distribution.

**Figure 4 sensors-22-01380-f004:**
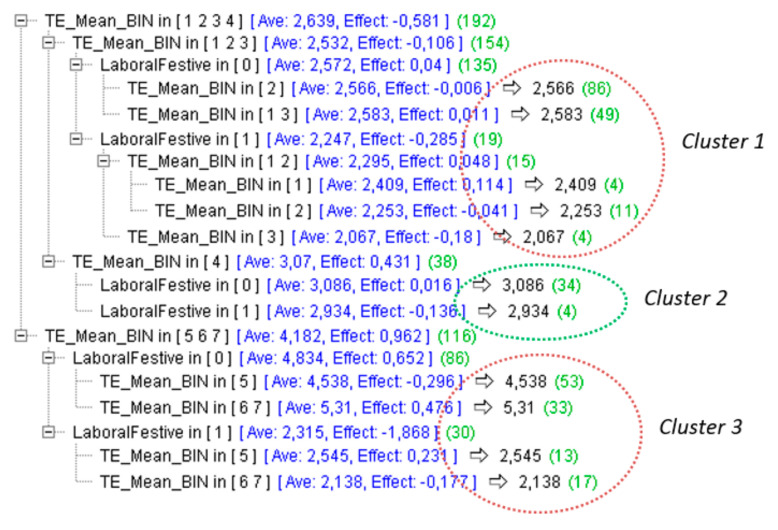
Set of rules for the clustering of the sample with the objective of average active power compressor versus temperature and work or non-workdays.

**Figure 5 sensors-22-01380-f005:**
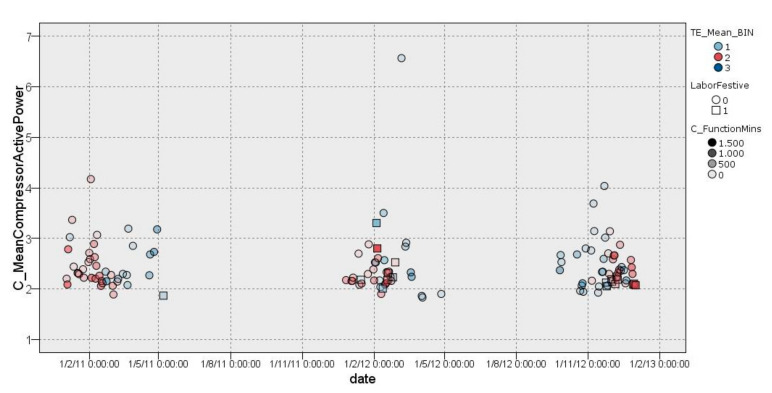
Graphical representation of distribution of cluster 1.

**Figure 6 sensors-22-01380-f006:**
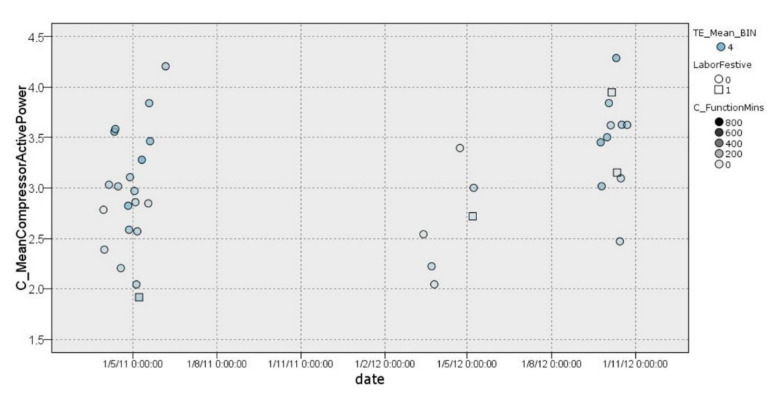
Graphical representation of distribution of cluster 2.

**Figure 7 sensors-22-01380-f007:**
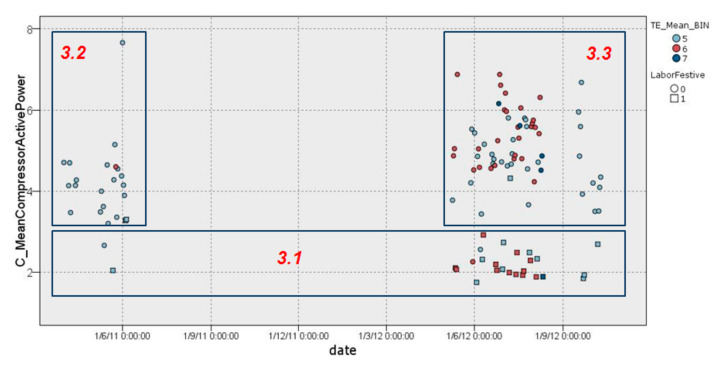
Graphical representation of distribution of cluster 3.

**Figure 8 sensors-22-01380-f008:**
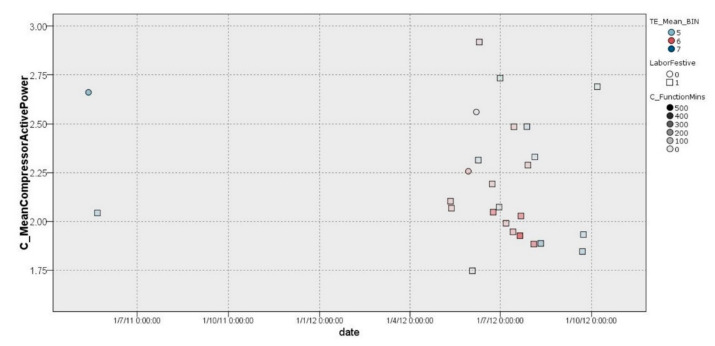
Graphical representation of distribution of cluster 3.1.

**Figure 9 sensors-22-01380-f009:**
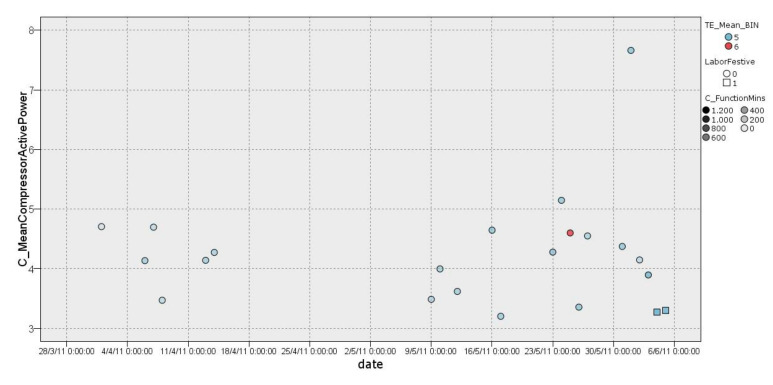
Graphical representation of distribution of cluster 3.2.

**Figure 10 sensors-22-01380-f010:**
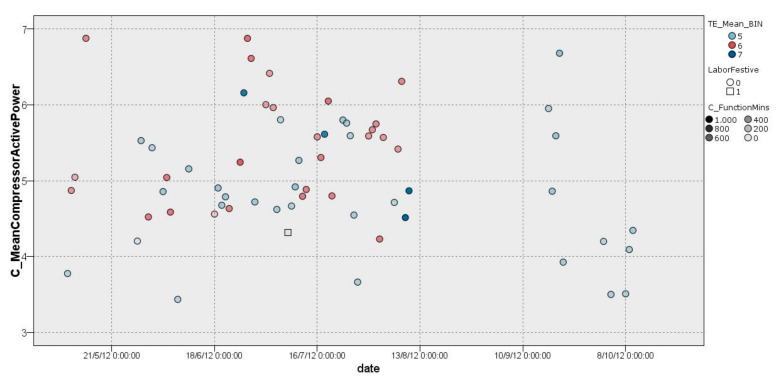
Graphical representation of distribution of cluster 3.3.

**Figure 11 sensors-22-01380-f011:**
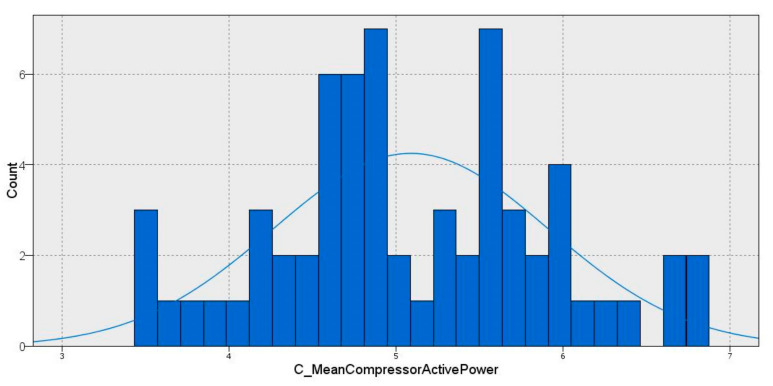
Gaussian distribution of energy consumption in cluster 3.3.

**Figure 12 sensors-22-01380-f012:**
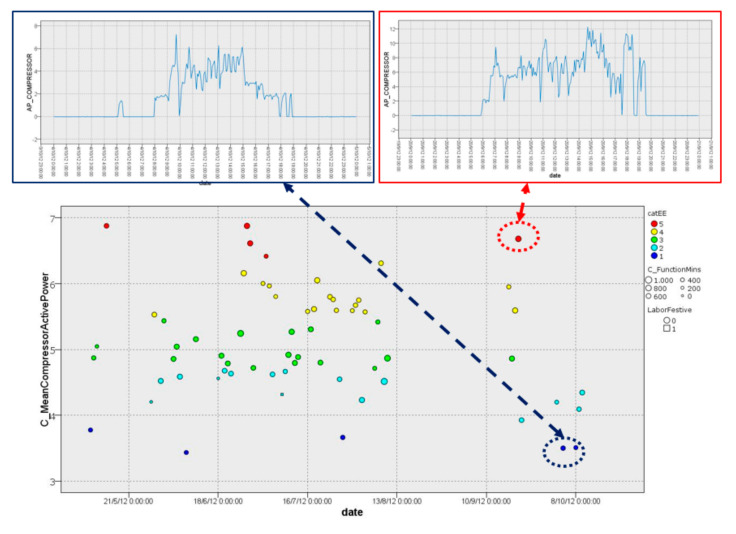
EE category prediction for cluster 3.3 and energy consumption behaviors.

**Table 1 sensors-22-01380-t001:** Selected features.

Feature	Unit
AEMET_AT	°C
AP_COMPRESSOR	kW
AP_COMPRESSOR_MINUTES	minutes
AP_COMPRESSORS_MEAN	kW
AP_LIGTH	kW
DATE	DD/MM/YY hh:min:ss
HOLIDAY	DD/MM/YY
LIG_MINUTES	minutes
NPSONC_MINUTES	minutes
NPSONC_PERIODNUMBER	integer
NPSONC_TIMES	integer
OCC_STOPPOINTS	integer
ORCC_PERIODNUMBER	integer
ORCC_MINUTES	minutes
ORCC_TIMES	integer
PRESENCE_ID_ENT	DD/MM/YY hh:min:ss
PRESENCE_ID_EXI	DD/MM/YY hh:min:ss
SOFFCCIT_MINUTES	minutes
SOFFCCIT_PERIODNUMBER	integer
SOFFCCIT_TIMES	integer
SONCCIT_AP	kW
SONCCIT_MINUTES	minutes
SONCCIT_PERIODNUMBER	integer
SONCCIT_TIMES	integer

**Table 2 sensors-22-01380-t002:** External temperature binning.

Bin	Lower	Upper
1	≥3,562,762	<8,562,762
2	≥8,562,762	<13,562,762
3	≥13,562,762	<18,562,762
4	≥18,562,762	<23,562,762
5	≥23,562,762	<28,562,762
6	≥28,562,762	<33,562,762
7	≥33,562,762	<38,562,762
8	≥38,562,762	<39,529,412

**Table 3 sensors-22-01380-t003:** Cluster 1 anomaly distribution. Each EE Category is associated with a color as depicted in Figure 12.

	Cluster 1	Days	OCC	ORCC	SOFFCCIT	SONCCIT	NPSONC	TOTAL
	*EE Categ. 1*	0	-	-	-	-	-	** *0* **
	*EE Categ. 2*	9	0	1	1	1	9	** *12* **
	*EE Categ. 3*	8	0	5	9	6	8	** *28* **
	*EE Categ. 4*	2	0	1	1	0	2	** *4* **
	*EE Categ. 5*	0	0	1	0	0	0	** *1* **
		** *19* **	** *0* **	** *8* **	** *11* **	** *7* **	** *19* **	** *45* **

**Table 4 sensors-22-01380-t004:** Cluster 2 anomaly distribution. Each EE Category is associated with a color as depicted in Figure 12.

	Cluster 1	Days	OCC	ORCC	SOFFCCIT	SONCCIT	NPSONC	TOTAL
	*EE Categ. 1*	3	0	0	0	0	1	** *1* **
	*EE Categ. 2*	8	0	0	0	0	1	** *1* **
	*EE Categ. 3*	13	0	0	0	0	3	** *3* **
	*EE Categ. 4*	12	0	2	0	0	0	** *2* **
	*EE Categ. 5*	2	0	2	0	0	1	** *3* **
		** *38* **	** *0* **	** *4* **	** *0* **	** *0* **	** *6* **	** *10* **

**Table 5 sensors-22-01380-t005:** Cluster 3.1 anomaly distribution. Each EE Category is associated with a color as depicted in Figure 12.

	Cluster 1	Days	OCC	ORCC	SOFFCCIT	SONCCIT	NPSONC	TOTAL
	*EE Categ. 1*	0	-	-	-	-	-	** *0* **
	*EE Categ. 2*	11	0	1	0	0	7	** *8* **
	*EE Categ. 3*	8	0	1	0	0	4	** *5* **
	*EE Categ. 4*	4	0	0	0	0	2	** *2* **
	*EE Categ. 5*	3	0	0	0	0	1	** *1* **
		** *26* **	** *0* **	** *2* **	** *0* **	** *0* **	** *14* **	** *16* **

**Table 6 sensors-22-01380-t006:** Cluster 3.2 anomaly distribution. Each EE Category is associated with a color as depicted in Figure 12.

	Cluster 1	Days	OCC	ORCC	SOFFCCIT	SONCCIT	NPSONC	TOTAL
	*EE Categ. 1*	0	-	-	-	-	-	** *0* **
	*EE Categ. 2*	7	0	3	0	0	3	** *6* **
	*EE Categ. 3*	12	0	3	0	0	3	** *6* **
	*EE Categ. 4*	2	0	0	0	0	0	** *0* **
	*EE Categ. 5*	1	0	1	0	0	0	** *1* **
		** *22* **	** *0* **	** *7* **	** *0* **	** *0* **	** *6* **	** *13* **

**Table 7 sensors-22-01380-t007:** Cluster 3.3 anomaly distribution. Each EE Category is associated with a color as depicted in Figure 12.

	Cluster 1	Days	OCC	ORCC	SOFFCCIT	SONCCIT	NPSONC	TOTAL
	*EE Categ. 1*	5	0	0	0	0	0	**0**
	*EE Categ. 2*	16	8	12	0	0	1	**21**
	*EE Categ. 3*	20	13	16	0	0	1	**30**
	*EE Categ. 4*	18	14	14	0	0	2	**30**
	*EE Categ. 5*	5	4	5	0	0	0	**9**
		** *64* **	** *39* **	** *47* **	** *0* **	** *0* **	** *4* **	** *90* **

**Table 8 sensors-22-01380-t008:** Stop points and minutes for OCC and ORCC anomalies vs. EE category. Each EE Category is associated with a color as depicted in Figure 12.

Cluster	Anomaly	Days	Categ. 1	Categ. 2	Categ. 3	Categ. 4	Categ. 5
*Cluster 1*	OCC	5	0	0	0	0	0
	ORCC	16	0	313 (1)	1769 (5)	418 (1)	544 (1)
*Cluster 2*	OCC	20	0	0	0	0	0
	ORCC	18	0	0	0	724 (2)	658 (2)
*Cluster 3.1*	OCC	20	0	0	0	0	0
	ORCC	18	0	340 (1)	315 (1)	0	0
*Cluster 3.2*	OCC	20	0	0	0	0	0
	ORCC	18	0	1020 (3)	945 (3)	0	543 (1)
*Cluster 3.3*	OCC	20	0	64 (8)	142 (13)	158 (14)	39 (4)
	ORCC	18	0	5026 (12)	7278 (16)	5881 (14)	2671 (5)
** *Total* **		** *64* **	** *0* **	** *64/6699* **	** *142/10307* **	** *158/7023* **	** *39/4416* **

**Table 9 sensors-22-01380-t009:** Average stop points and minutes per day for OCC and ORCC anomalies. Each EE Category is associated with a color as depicted in Figure 12.

Cluster	Anomaly	Days	Categ. 1	Categ. 2	Categ. 3	Categ. 4	Categ. 5
*Cluster 1*	OCC	5	0	0	0	0	0
	ORCC	16	0	313	353.8	418	544
*Cluster 2*	OCC	20	0	0	0	0	0
	ORCC	18	0	0	0	362	329
*Cluster 3.1*	OCC	20	0	0	0	0	0
	ORCC	18	0	340	315	0	0
*Cluster 3.2*	OCC	20	0	0	0	0	0
	ORCC	18	0	462.7	437.7	0	543
*Cluster 3.3*	OCC	20	0	8	10.9	11.3	9.8
	ORCC	18	0	418.8	454.9	420.1	534.2
** *Total* **		** *64* **	** *0* **	** *8/394.1* **	** *10.9/412.3* **	** *11.3/413.1* **	** *9.8/490.7* **

**Table 10 sensors-22-01380-t010:** Distribution of days for each cluster based on EE category. Each EE Category is associated with a color as depicted in Figure 12.

Cluster	Days	Categ. 1	Categ. 2	Categ. 3	Categ. 4	Categ. 5
*Cluster 1*	171	0%	32.35%	48.87%	13.53%	5.25%
*Cluster 2*	91	7.89%	21.05%	34.21%	31.58%	5.27%
*Cluster 3.1*	26	0%	42.31%	30.77%	15.39%	11.53%
*Cluster 3.2*	22	0%	31.82%	54.55%	9.09%	4.54%
*Cluster 3.3*	64	7.81%	25%	31.25%	28.13%	7.81%
** *Total* **	** *374* **	** *3.25%* **	** *29%* **	** *41.36%* **	** *20.29%* **	** *6.36%* **

## Data Availability

All data and code will be made available on request to the correspondent author’s email with appropriate justification.

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
