# Peer review of "Decision Support System to Classify and Optimize the Energy Efficiency in Smart Buildings: A Data Analytics Approach"

_sensors, 2022, doi:10.3390/s22041380_

Round 1

Reviewer 1 Report

In this contribution, a decision support system has been proposed with the aim to optimize the energy efficiency in smart buildings.

Overall, it is an interesting and well-written paper. However, I have the following concerns which should be addressed.

In such a paper, focused on the task of optimizing the energy efficiency in smart building, it is useful to the readers to outline the main technologies employed for the data acquisition. You can find two review articles, hereafter:  doi.org/10.1063/5.0044673 - 10.1016/j.enbuild.2015.02.028.

How do the sensors in BlueNet building detect the presence of people? I suggest to add this information for the readers.

Figure 1 is missing, only the caption is visible. Having a graphical representation of the architecture followed for the EE optimization system might be an important element for reviewing the manuscript

Figure 2 should not be a figure but a table. Moreover, it was added as a low-quality image.

Also the quality of Figure 5 should be enhanced.

Is the proposed methodology applicable only to the BlueNet building or the proposed model is applicable to different environments? In the second case, what strategies should be undertaken?

The novelty of this paper compared to the existing literature should be highlighted.

Author Response

Dear reviewer, 

First at all, thank you for your comments and suggestions. 

Please see the attachment for the comments and modifications.

With kind regards,

The authors.

Reviewer 2 Report

The paper is well written and structured, the references list is well though and supports the paper's findings in great extend. The topic is of course of great interest. A few comments I would like to add:

  1. I would like to see - if possible - some kind of benchmarking of the proposed solution/implementation against other pre-published research. Why does this solution outperform previous ones? Is it more accurate, does it take more parameters into account?
  2. For such a scheme, I would expect energy consumption data from (but not necessarily) from smart meters to be considered as input as well. That would help even for for an energy disaggregation layer to further pinpoint activities/appliances etc that need specific treatment. Is that something the authors would like to include, as future work maybe?
  3. Figure 1, regarding the arhitecture is not printed in my .pdf I cannot see it. All the other figures are clear.

Author Response

(The authors gave the same response as above.)
